# A *COL5A2* In-Frame Deletion in a Chihuahua with Ehlers-Danlos Syndrome

**DOI:** 10.3390/genes13050934

**Published:** 2022-05-23

**Authors:** Sarah Kiener, Lucie Chevallier, Vidhya Jagannathan, Amaury Briand, Noëlle Cochet-Faivre, Edouard Reyes-Gomez, Tosso Leeb

**Affiliations:** 1Institute of Genetics, Vetsuisse Faculty, University of Bern, 3001 Bern, Switzerland; sarah.kiener@vetsuisse.unibe.ch (S.K.); vidhya.jagannathan@vetsuisse.unibe.ch (V.J.); 2Dermfocus, University of Bern, 3001 Bern, Switzerland; 3“Biology of the Neuromuscular System” Team, INSERM, IMRB, Univ Paris-Est Créteil, Ecole Nationale Vétérinaire d’Alfort, 94700 Maisons-Alfort, France; lucie.chevallier@vet-alfort.fr (L.C.); edouard.reyes-gomez@vet-alfort.fr (E.R.-G.); 4Unité de Dermatologie, ChuvA, Ecole Nationale Vétérinaire d’Alfort, 94700 Maisons-Alfort, France; amaury.briand@vet-alfort.fr (A.B.); noelle.cochet-faivre@vet-alfort.fr (N.C.-F.); 5Unité d’Histologie et d’Anatomie Pathologique, BioPôle Alfort, Ecole Nationale Vétérinaire d’Alfort, 94700 Maisons-Alfort, France

**Keywords:** *Canis lupus familiaris*, dog, dermatology, genodermatosis, skin, precision medicine, animal model

## Abstract

Ehlers-Danlos syndrome (EDS) is a group of heterogeneous, rare diseases affecting the connective tissues. The main clinical signs of EDS are skin hyperextensibility, joint hypermobility, and skin fragility. Currently, the classification of EDS in humans distinguishes 13 clinical subtypes associated with variants in 20 different genes, reflecting the heterogeneity of this set of diseases. At present, variants in three of these genes have also been identified in dogs affected by EDS. The purpose of this study was to characterize the clinical and histopathological phenotype of an EDS-affected Chihuahua and to identify the causative genetic variant for the disease. The clinical examination suggested a diagnosis of classical EDS. Skin histopathology revealed an abnormally thin dermis, which is compatible with classical EDS. Whole-genome sequencing identified a heterozygous de novo 27 bp deletion in the *COL5A2* gene, *COL5A2*:c.3388_3414del. The in-frame deletion is predicted to remove 9 amino acids in the triple-helical region of COL5A2. The molecular analysis and identification of a likely pathogenic variant in *COL5A2* confirmed the subtype as a form of classical EDS. This is the first report of a *COL5A2*-related EDS in a dog.

## 1. Introduction

Ehlers-Danlos syndrome (EDS) is a clinically and genetically heterogenous group of heritable connective tissue disorders. Typical clinical signs include joint hypermobility, skin hyperextensibility, and tissue fragility [1]. Different types of EDS can be distinguished according to the underlying pathomechanisms, such as defective primary structure and processing of collagen, collagen folding and cross-linking, structure and function of the myomatrix, glycosaminoglycan biosynthesis, intracellular processes, or complement pathway [2]. In humans, variants in 20 genes have so far been shown to cause different forms of EDS (Table 1) [2].

In dogs, EDS was first described as cutaneous asthenia more than 70 years ago [3]. To date, several reports of dogs with connective tissue disorders, such as EDS have been published [4,5,6,7,8,9,10,11,12,13,14,15,16,17]. However, only for a few of them, the underlying genetic variant has been identified. EDS in a Doberman Pinscher was caused by a homozygous nonsense variant in *ADAMTS2* (OMIA 000328-9615) [18], which encodes the procollagen I N-proteinase that excises the N-propeptide of type I and type II procollagens [19].

Two compound heterozygous missense variants in *TNXB* were reported in a single mixed-breed dog with EDS. However, the evidence for pathogenicity was extremely weak and it is not fully clear whether this dog really suffered from a *TNXB*-related form of EDS (OMIA 002203-9615) [20]. The encoded tenascin-X is a large extracellular matrix protein, which is an essential regulator of collagen deposition by dermal fibroblasts [21,22,23].

Two independent variants in *COL5A1* have been identified in a Labrador Retriever and a mixed-breed dog affected with EDS (OMIA 002165-9615) [24]. *COL5A1* encodes the α1 chain of type V collagen, which is important for correct collagen fibrillogenesis [25].

In this study, we investigated the clinical and histopathological phenotype of an EDS-affected Chihuahua and the underlying causative genetic defect.

## 2. Materials and Methods

### 2.1. Animal Selection

This study included an EDS-affected Chihuahua and its unaffected parents. For the whole genome sequencing data analysis, we used 783 control dogs from different breeds and 9 wolves (Appendix A). The control dogs and wolves had already been used in earlier studies, e.g., [26,27].

### 2.2. Clinical and Histopathological Examinations

The affected dog and both its parents underwent clinical examination. A piece of skin flap from a trauma-induced wound at the dorsum of the affected dog was fixed in 10% neutral-buffered formalin, processed routinely, and sections were stained with hematoxylin-eosin and saffron (HES).

### 2.3. DNA Extraction

Genomic DNA was isolated from EDTA blood of the affected dog using the Maxwell^®^ 16 Blood DNA Purification Kit and from buccal cells of both parents using the Maxwell^®^ 16 Buccal Swab LEV DNA Purification Kit, both using the Maxwell^®^ 16 Instrument (Promega, Dübendorf, Switzerland). Genomic DNA was frozen at −20 °C until further use.

### 2.4. Whole-Genome Sequencing and Variant Calling

An Illumina TruSeq PCR-free DNA library with ~340 bp insert size of the affected dog was prepared and sequenced on a NovaSeq 6000 instrument with 25× coverage (Illumina, San Diego, CA, USA). The sequence data were submitted to the European Nucleotide Archive with the study accession PRJEB16012 and the sample accession SAMEA8797073. Mapping, alignment, and variant calling were performed as described [26]. Private variant filtering was performed with a hard filtering approach requiring the genotype 0/1 for heterozygous or 1/1 for homozygous variants in the affected dog and simultaneously a homozygous reference or missing genotype in the control dogs (0/0 or ./.).

### 2.5. Gene Analysis

All references within the canine *COL5A2* gene correspond to the NCBI RefSeq accession numbers XM_005640393.3 (mRNA) and XP_005640450.1 (protein). We used the CanFam3.1 reference genome assembly and NCBI annotation release 105.

### 2.6. PCR and Sanger Sequencing

Sanger sequencing was used to validate the candidate variant *COL5A2*:c.3388_3414del in the affected dog and to genotype both parents. PCR products were amplified from 10 ng genomic DNA using GoTaq^®^ G2 Flexi DNA Polymerase in a total volume of 25 µL including 0.2 µL *Taq* polymerase, 5 µL 5× buffer, 1.5 µL MgCl_2_ solution at 25 mM (Promega, Madison, WI, USA), 0.5 µL dNTP mix at 10 mM each (MP Biomedicals, Irvine, CA, USA) together with 0.5 µL each forward primer 5′-TAGCGTTCAGGCTTCCACTG-3′ and reverse primer 5′-CTCCAACACCTACGTGAGCC-3′ (primer concentrations were 10 µM). PCR amplification comprised 32 cycles (denaturation 30 s at 94 °C, annealing 40 s at 60 °C, and elongation 40 s at 72 °C) followed by 5 min at 72 °C. Electrophoresis was performed on a 2% agarose gel. After DNA gel extraction using the QIAquick Gel Extraction Kit (QIAGEN, Hilden, Germany), amplicons were sequenced on an ABI 3730XL DNA Analyzer (Thermo Fisher Scientific, Reinach, Switzerland). Sanger sequences were analyzed using the Chromas 2.6.6 software (Technelysium, Pty, Ltd., South Brisbane, Australia).

## 3. Results

### 3.1. Clinical History and Examination

An 18-month-old spayed female Chihuahua was referred to the dermatology consultation for suspicion of EDS. The first clinical signs were noted on both eyes during puppyhood, at which time the dog was presented to the ophthalmologist with a diagnosis of corneal endothelial dystrophy. Moreover, its skin was hyperextensible (Figure 1a) and abnormalities were noted on the face around the eyes and on the extremities where the skin was fragile and tore easily after scratching or rubbing (Figure 1b–d). Several episodes of wounds on the trunk after minor injuries or scratching were also reported (Figure 1e). The healing process was difficult and long at each time. The dog had presented bilateral inguinal hernia. The dog lived with both parents and its two sisters and was the only one presenting these abnormalities. The dog wore a coat and a protective pet cone all day long in order to protect the skin and prevent any injuries and wound formation (Figure 2). At the time of consultation, the dog was in good general health condition. The skin was hyperextensible and no wounds were noted during clinical examination.

### 3.2. Histopathological Examination

Histopathological examination showed a marked reduction in dermis thickness (Figure 3a). The density of collagen fibers was markedly reduced giving a loose appearance of the dermis. There was moderate variation in collagen fibers size (Figure 3b). The epidermis and hypodermis were unremarkable.

### 3.3. Genetic Analysis

Whole-genome sequencing and private variant filtering were performed to identify the causative genetic variant. For this analysis, we compared the sequence data of the affected dog to the genomes of 783 control dogs of different breeds and 9 wolves (Appendix A). We specifically looked for protein-changing variants in the 20 known EDS candidate genes (Table 1). The outcome of the various filtering steps is summarized in Table 2 and Appendix A.

We identified a single private protein-changing variant within an EDS candidate gene, *COL5A2*. The variant can be designated as Chr36:30,548,697_30,548,723del. It is a heterozygous in-frame deletion of 27 bases (XM_005640393.3:c.3388_3414del), predicted to remove 9 amino acids, XP_005640450.1:p.(Lys1130_Asp1138del). The presence of the variant in a heterozygous state in the EDS-affected Chihuahua was confirmed by Sanger sequencing. The genotyping results of both parents revealed that *COL5A2*:c.3388_3414del represented a de novo variant as the mutant allele was absent from buccal cell DNA of both parents.

## 4. Discussion

In this study, we describe a Chihuahua with Ehlers-Danlos syndrome. EDS is a heterogeneous group of diseases affecting connective tissue. In human patients, 13 clinically different EDS subtypes with known causal variants in 20 different genes are recognized. A recently introduced additional clinical subtype has not yet been characterized at the molecular level [2].

In humans, the diagnosis of an EDS subtype is made by identifying clinical signs that can be categorized into major or minor criteria and by complementary molecular analysis. For the so-called classical EDS, the major and minor criteria are listed in Table 3.

Minimal criteria suggestive for classical EDS are the presence of either both major criterions or one major criterion and at least three minor criteria [1]. As this classification of EDS subtypes has been developed in humans, it must be used with caution in animals. Nevertheless, given the clinical, histopathological, and molecular similarities between animal models of EDS and human EDS, it is realistic to use it to define an animal EDS subtype.

The dog described here presented in its clinical history the first major criterion, namely (1) hyperextensibility of the skin and atrophic scarring, as well as three clinical signs listed as minor criteria, namely (1) easy bruising, (3) skin fragility and traumatic splitting, and (6) hernia. Thus, by referring to the human classification, the dog falls into the clinical category of classical EDS. As in humans, clinical diagnosis must be supplemented by molecular analysis. In our case, the genetic analysis confirmed the clinical hypothesis of a classical EDS.

Suspicion of EDS in animals usually leads to performing a skin biopsy and histopathological analysis of the dermis. Classically, histopathological descriptions report a dermis of normal thickness but present disorganized, smaller, curved collagen fibers of variable length and unequal diameter [9,16,24,28,29]. However, these characteristics are not always found and some affected animals show no abnormality on histological analysis. In some cases, a dermis of reduced thickness has been identified in affected animals, notably cats and dogs [5,29], as it is in our case.

Using a whole-genome sequencing approach, we identified a single heterozygous protein-changing variant in a known EDS candidate gene in the investigated EDS-affected Chihuahua. The variant is an in-frame deletion of 27 bases in *COL5A2*, encoding the α2 chain of type V collagen.

Type V collagen is present mainly as heterotrimers of two α1(V) chains (encoded by *COL5A1*) and one α2(V) chain [30,31]. Homotrimers of three α1(V) chains or heterotrimers comprised of an α1(V), α2(V), and α3(V) chain (encoded by *COL5A3*) also exist; however, their physiological function is largely unknown [2,32]. Type V collagen co-assembles with type I collagen into heterotypic fibrils in the extracellular matrix, with type V collagen being crucial for initial fibril formation [25]. Correct fibril formation and integrity play a key role in maintaining the physical properties of skin and other tissues [2].

The three chains of type V collagen are assembled into a triple helix. The sequence of each procollagen chain is characterized by extended Gly-Xaa-Yaa repeats. The presence of glycine (which has no side chain) in every third position permits the formation of the triple-helical structure. The Xaa and Yaa are often proline and hydroxyproline but can be any amino acid [2].

*COL5A1*-associated EDS mostly results from *COL5A1* haploinsufficiency, as type V procollagen molecules cannot accommodate more than a single proα2(V) chain, and the reduction of available proα1(V) chains results in the production of about half the normal amount of type V collagen [33]. By contrast, proα1(V) chains can form functional homotrimers [34]. Thus, no heterozygous *COL5A2* null alleles have been identified in EDS patients so far [35].

*COL5A2*-associated EDS is mostly related to structural variants located in the triple helix domain, resulting in the production of mutant proα2(V) chains which are expected to be incorporated in defective type V collagen molecules [35].

In accordance with [35], the *COL5A2* variant identified in the EDS-affected dog from this study is predicted to remove three Gly-Xaa-Yaa triplet repeats and thus induce a structural alteration of the synthesized mutant proα2(V) chains. We assume that this causes aberrant heterotrimer formation and a defective type V collagen structure corresponding to a dominant-negative gain-of-function in the mutant allele.

In domestic animals, only one other *COL5A2* variant, p.Gly789Val, has been reported to cause EDS in Holstein cattle [28].

## 5. Conclusions

To the best of our knowledge, our study represents the first report of a *COL5A2*-related EDS in dogs. The reported *COL5A2* deletion arose by a de novo mutation event, which strongly supports its causality for the EDS phenotype.

## Figures and Tables

**Figure 1 genes-13-00934-f001:**
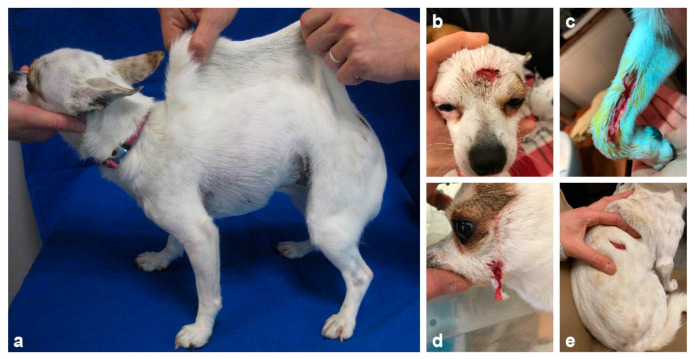
Clinical signs of the classical Ehlers-Danlos-affected Chihuahua dog. (**a**) Skin hyperextensibility over the dorsum. (**b**) Skin laceration resulting from minimal trauma on the head, (**c**) on the forelimb, (**d**) on the lower jaw, and (**e**) on the dorsum.

**Figure 2 genes-13-00934-f002:**
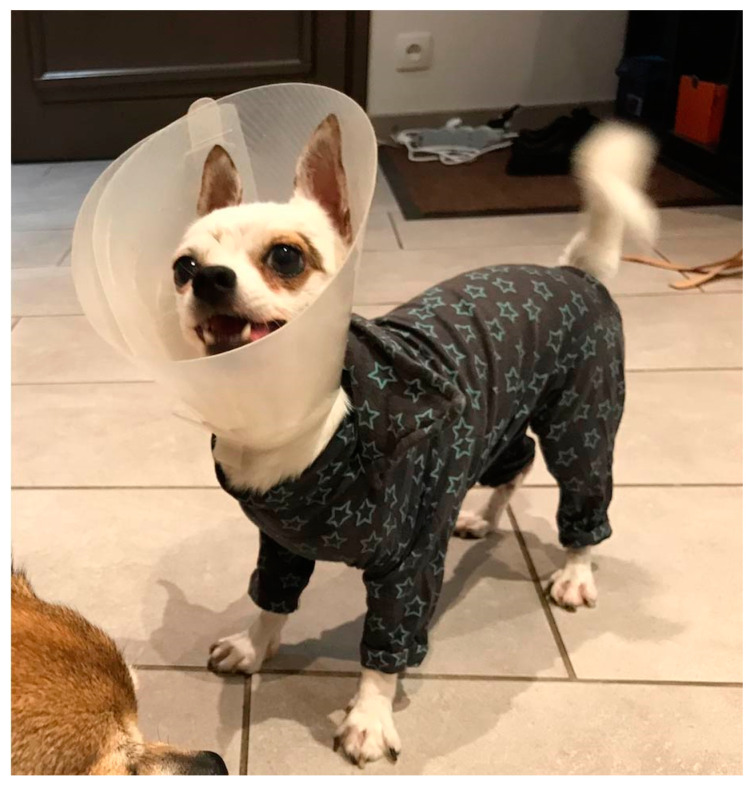
The dog lives continuously with a protective suit and an Elizabethan collar to prevent wound formation.

**Figure 3 genes-13-00934-f003:**
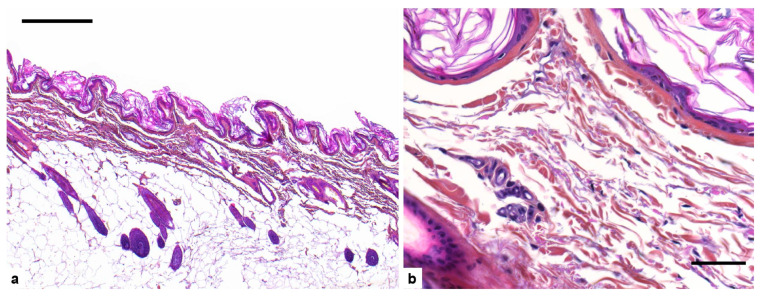
Cutaneous histopathological findings of the dog affected with Ehlers-Danlos syndrome (EDS). (**a**) Dermis is abnormally thin while epidermis and hypodermis are normal. Hematoxylin-eosin and saffron stain (HES), bar 500 µm; (**b**) collagen fibers are moderately uneven in size and their density is markedly reduced. Hematoxylin-eosin and saffron stain (HES), bar 50 µm.

**Table 1 genes-13-00934-t001:** Overview on genetic causes of human Ehlers-Danlos syndrome (EDS), adapted from [2].

Gene	EDS Type	Inheritance
*ADAMTS2*	Dermatosparaxis	AR
*AEBP1*	Classical-like type 2	AR
*B3GALT6*	Spondylo-dysplastic	AR
*B4GALT7*	Spondylo-dysplastic	AD
*CHST14*	Musculocontractural	AR
*COL1A1*	Classical, Vascular, Artrochalasia	AD
*COL1A2*	Artrochalasia, Cardiac valvular	AD
*COL3A1*	Vascular	AD
*COL5A1*	Classical	AD
*COL5A2*	Classical	AD
*COL12A1*	Myopathic	AD
*C1R*	Periodontal	AD
*C1S*	Periodontal	AD
*DSE*	Musculocontractural	AR
*FKBP14*	Kyphoscoliotic	AR
*PRDM5*	Brittle cornea syndrome	AR
*PLOD1*	Kyphoscoliotic	AR
*SLC39A13*	Spondylo-dysplastic	AR
*TNXB*	Classical-like	AR
*ZNF469*	Brittle cornea syndrome	AR

**Table 2 genes-13-00934-t002:** Results of variant filtering in the affected dog against 792 control genomes.

Filtering Step	Heterozygous Variants	Homozygous Variants
All variants in the affected dog	4,191,411	2,457,482
Private variants	18,956	1258
Protein-changing private variants	104	2
Protein-changing private variants in 20 candidate genes	1	0

**Table 3 genes-13-00934-t003:** Classical EDS clinical criteria in humans, adapted from [1].

Major criteria	1. Skin hyperextensibility and atrophic scarring
	2. Generalized joint hypermobility
Minor criteria	1. Easy bruising
	2. Soft, doughy skin
	3. Skin fragility (or traumatic splitting)
	4. Molluscoid pseudotumors
	5. Subcutaneous spheroids
	6. Hernia (or history thereof)
	7. Epicanthal folds
	8. Complication of joint hypermobility (e.g., sprains, luxation/subluxation, pain, flexible flatfoot)
	9. Family history of a first degree relative who meets clinical criteria

## Data Availability

The accessions for the sequence data reported in this study are listed in Appendix A.

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
