# Peer review of "A COL5A2 In-Frame Deletion in a Chihuahua with Ehlers-Danlos Syndrome"

_genes, 2022, doi:10.3390/genes13050934_

Round 1

Reviewer 1 Report

The results of the research make a significant contribution to understanding the genetic basis of dermatological diseases in dogs. It is also important from the point of view of human diseases, because in research on their genetic conditioning and inheritance, the dog is more and more often used as a model animal.

Regarding the Materials and methods chapter, I have a few questions and comments:
- would it not be a good solution to include in the research also siblings of the tested individual, there is also no information whether there were cases of EDS among the siblings
- I would suggest supplementing the information on dogs included in the control group in the Material and Methods chapter, i.e. whether all dogs were analyzed as part of this project, or were they data from the database and deposited there as part of other projects
- if these were dogs in control group for which full genome sequences were obtained under this project, what material were the analyzes made of
- there is also no information as to whether the material for genetic testing has been frozen or whether the DNA was fresh and isolated after the material was collected
- there is no information in section 2.6 about the PCR reaction itself, for example the composition of the reaction mixture or its conditions.

Author Response

(1)

The results of the research make a significant contribution to understanding the genetic basis of dermatological diseases in dogs. It is also important from the point of view of human diseases, because in research on their genetic conditioning and inheritance, the dog is more and more often used as a model animal.

Response: Thank you very much for the favorable evaluation.

(2)

Regarding the Materials and methods chapter, I have a few questions and comments:

Would it not be a good solution to include in the research also siblings of the tested individual, there is also no information whether there were cases of EDS among the siblings.

Response: We stated in the results paragraph (3.1 Clinical history and examination, lines114/115): “The dog lived with both parents and its two sisters and was the only one presenting these abnormalities.“ As both healthy parents carried wild-type alleles at the COL5A2 locus, we did not genotype the two healthy sisters of the affected dog. (The pathogenic allele arose due to a de novo mutation event in the affected dog. It would not be expected to occur in any of the siblings.)

(3)

I would suggest supplementing the information on dogs included in the control group in the Material and Methods chapter, i.e. whether all dogs were analyzed as part of this project, or were they data from the database and deposited there as part of other projects

Response: The control dogs were derived from earlier projects and their genome sequence data are publicly available. We expanded the methods section to make this clear.

(4)

If these were dogs in control group for which full genome sequences were obtained under this project, what material were the analyzes made of

Response: All the control dogs were sequenced in earlier projects. For the vast majority of the control dogs, the DNA for genome sequencing was isolated from EDTA blood (peripheral blood mononuclear cells). For a very small number of control dogs and wolves, the DNA was isolated from hair roots or skin samples.

(5)

There is also no information as to whether the material for genetic testing has been frozen or whether the DNA was fresh and isolated after the material was collected

Response: We froze the DNA between isolation and subsequent experiments. This information was added to section 2.3.

(6)

There is no information in section 2.6 about the PCR reaction itself, for example the composition of the reaction mixture or its conditions.

Response: We added more detailed information on the PCR methodology to section 2.6.

Reviewer 2 Report

The paper is straightforward and results are clear.

Author Response

(1)

The paper is straightforward and results are clear.

Response: Thank you very much for the favorable evaluation.